# Rewriting Universes: Post-Brexit Futures in Dave Hutchinson's *Fractured Europe* Quartet

Hadas Elber-Aviram 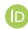

Department of English, The University of Notre Dame (USA) in England, London SW1Y 4HG, UK; helberav@nd.edu

**Abstract:** Recent years have witnessed the emergence of a new strand of British fiction that grapples with the causes and consequences of the United Kingdom's vote to leave the European Union. Building on Kristian Shaw's pioneering work in this new literary field, this article shifts the focus from literary fiction to science fiction. It analyzes Dave Hutchinson's *Fractured Europe* quartet—comprised of *Europe in Autumn* (pub. 2014), *Europe at Midnight* (pub. 2015), *Europe in Winter* (pub. 2016) and *Europe at Dawn* (pub. 2018)—as a case study in British science fiction's response to the recent nationalistic turn in the UK. This article draws on a bespoke interview with Hutchinson and frames its discussion within a range of theories and studies, especially the European hermeneutics of Hans-Georg Gadamer. It argues that the *Fractured Europe* quartet deploys science fiction *topoi* to interrogate and criticize the recent rise of English nationalism. It further contends that the *Fractured Europe* books respond to this nationalistic turn by setting forth an estranged vision of Europe and offering alternative modalities of European identity through the mediation of photography and the redemptive possibilities of cooking.

**Keywords:** speculative fiction; science fiction; utopia; post-utopia; dystopia; Brexit; England; Europe; Dave Hutchinson; *Fractured Europe* quartet





## 1. Introduction

In his 1982 essay, "Progress versus Utopia, or, Can We Imagine the Future?", Fredric Jameson observed that science fiction's visions of the future are fundamentally about the present moment:

> For the apparent realism, or representationality, of SF has concealed another, far more complex temporal structure: not to give us "images" of the future—whatever such images might mean for a reader who will necessarily predecease their "materialization"—but rather to defamiliarize and restructure our experience of our own *present*, and to do so in specific ways distinct from all other forms of defamiliarization. (Jameson [1982] 2005b, p. 286)

One of these specific ways in which science fiction defamiliarizes our experience of the present moment is through what Peter Stockwell has termed "alternativity" (Stockwell 1996, p. 5), "a metaphorical strategy whereby elements and processes from our apparent reality are worked out in another domain" (p. 5), often taking "the historical dimension to extrapolate from current realities into the future" (p. 5). Stockwell claims that "the base" reality from which science fiction extrapolates its futures is recognizable through clues embedded in the science fiction text (p. 5). However, as Elana Gomel (2010) has observed, this assumption of a consensus reality does not cohere with the present moment: "the nature of the world we live in is a fiercely contested issue, the arena of both metaphorical and real battles" (p. 14). Zygmunt Bauman (2017) has diagnosed a "new wave of tribalism" that has recently exacerbated the fragmentation of present-day society into myriad epistemologies that are becoming increasingly insulated from one another (p. 52). This disintegration is by no means a new phenomenon—"the fragmentation of

grand narratives into little narratives" has been theorized as the distinctive feature of postmodernism since the 1970s if not earlier (Currie 1998, p. 107). However, the brazenness of its political instrumentalization and the scale of its real-life consequences in the era of Brexit, Trump and COVID-19 has culminated in a worldwide crisis. Science fiction has a key role to play at this time of increasing nationalism, jingoism, isolationism, intolerance, and violence, which are fueled by conflicting assumptions about reality, history, facts and truth. The genre's repertoire of alien encounters, non-linear temporalities and superimposed parallel worlds render it uniquely equipped to respond to our current *zeitgeist*.

Robert Eaglestone's ground-breaking volume of essays, *Brexit and Literature: Critical and Cultural Responses* (Eaglestone 2018), explores the role of English literature in responding to, and making sense of, the political crisis triggered by the United Kingdom's vote to leave the European Union in the 2016 EU referendum. Kristian Shaw's chapter in that volume merits special note, as it coins the term "BrexLit" to describe "fictions that either directly respond or imaginatively allude to Britain's exit from the EU" (Shaw 2018, p. 18). Shaw focuses on literary fiction written in the referendum's immediate aftermath, but he expands the scope of his discussion to include pre-Brexit novels that engage with similar themes.

This article builds on Shaw's pioneering work and follows his lead in including pre-Brexit fictions within its purview, but it shifts the focus from literary fiction to science fiction. It analyzes Dave Hutchinson's *Fractured Europe* quartet—comprised of *Europe in Autumn* (Hutchinson 2014), *Europe at Midnight* (Hutchinson 2015), *Europe in Winter* (Hutchinson 2016) and *Europe at Dawn* (Hutchinson 2018)—as a case study in British science fiction's response to the recent nationalistic turn in the UK. It draws on a range of theoretical frameworks, especially the European hermeneutics of the German philosopher Hans-Georg Gadamer, together with a three-hour interview with Dave Hutchinson conducted especially for this purpose. This article argues that the *Fractured Europe* quartet deploys the science fiction *topoi* of alternate worlds, immersive virtual environments, and near-future settings, as well as literary allusions and manipulations of narrative chronology, to interrogate and criticize the rise of English nationalism in recent years. It further contends that the *Fractured Europe* books respond to this nationalistic turn by setting forth an estranged and enticing vision of Europe as seen through the eyes of an Englishman from another world. The books invest their utopian energies in a fictional trans-European railway line and offer alternative modalities of European identity through the mediation of photography and the redemptive possibilities of cooking.

## 2. Brexit Fantasies, Science Fiction Alternatives

The focus of this article is not Brexit itself, neither its causes nor its consequences, but rather its representations in the science fiction genre. Yet, a brief discussion of Brexit and the complicated and self-contradictory "structure of feeling" that it expressed is nevertheless warranted for contextualization of this subject matter (Williams 1961, p. 48). As Fintan O'Toole argues in his illuminating study, *Heroic Failure: Brexit and the Politics of Pain* (O'Toole 2018), Brexit was the political corollary of a longstanding collective British fantasy of victimhood that recast the United Kingdom as an invaded colony rather than a former imperial power. O'Toole contends that British alternate histories, a subgenre of British science fiction, provide distilled expressions of this fantasy by imagining a scenario in which Britain lost the Second World War and was conquered by the Third Reich. Len Deighton's *SS-GB* (Deighton [1978] 2009), not incidentally one of Dave Hutchinson's favorite Deighton novels (see Hutchinson 2021, n.p.), and Robert Harris's *Fatherland* (Harris [1992] 2009) both enact Britain's "vertiginous fall from 'heart of Empire' to 'occupied colony'" (O'Toole 2018, p. 26). These alternate histories thereby express, in fictionalized form, British frustrations with the postwar decline of the United Kingdom together with Britons' collective need to reconcile the contradictions in British identity that emerge from their "self-image as exemplars of liberty and civility and the violence and domination that were the realities of Empire" (p. 45). Thus, these fictions of alternate history afford British readers the distinct

pleasure of a once-colonizing nation "imagining itself as the colonized" (p. 42), their guilt at reaping the lasting fruits of Empire assuaged in fantasies ranging from resigned submission to heroic resistance against an oppressive superstate.

As O'Toole has shown, the Brexit campaign tapped into this deep-seated fantasy while substituting the EU for the Third Reich as the imperial oppressor in this daydream of victimhood: "It does not greatly matter what the European Union is or what it is doing—its function in the plot is to be a more insidious form of Nazism" that has conquered Britain by cunning and deceit (p. 34). This fantasy allowed Brexiteers to cast their campaign in the heroic mold of an "imaginary existential struggle between the gallant English Resistance and the Euroreich" (p. 34). O'Toole's argument sheds light on the science fictionality of Brexit, its thorough grounding in a counterfactual understanding of the world and its sublimation of a premise that has been deployed in actual works of science fiction, namely that Britain had lost the Second World War, suffered conquest and colonization, endured heroic suffering, and launched a plucky resistance against the European oppressor. Consequently, the science fiction genre is particularly well suited to respond to the ideological challenges posed by Brexit by critiquing it on its home turf, using the generic topoi of alternate histories in the service of discourses that resist and problematize simplified conceptualizations of European history, real or imagined.

In *Europe in Winter*, the protagonists discover computer processing power capable of "running all possible scenarios of European history" (Hutchinson 2016, p. 33), attempting to "predict even the dozen or so most likely outcomes of an action" (p. 36). These programs create simulations of the protagonist Rudi—an Estonian chef turned smuggler—two of whom encounter each other and discuss the differences between their respective virtual worlds. Their conversation is particularly striking for the conspicuous absence of any reference to England or Britain, thereby subtly demolishing the Leave campaign's overinflated sense of the importance of Britain in the European narrative. The two Rudis discuss questions such as "Does the European Union still exist?" and "Do you have *The Matrix* here?" (pp. 32, 30), but Britain is of no interest. It is neither the plucky underdog fighting for liberty nor a power to be reckoned with, but simply irrelevant to a discussion of their branching realities of everyday life in Europe. Indeed, in the alternate world outside of these computer simulations, England is first mentioned in passing as one of the last countries to retain its membership in a collapsing EU, its position on the global stage likened to that of a branch of "Burger King" in a world that has lost its taste for fast food (Hutchinson 2014, p. 27).

This alternate Europe and its virtual counterparts all neatly sidestep identification with our own world—one of them depicts a near-future Europe resembling our own, but it is a world without *Star Trek*—thereby suggesting an infinite variety of permutations beyond the ones presented in the books. This conceit presupposes what Gomel has parsed as "a fluid contingency, in which a single cause might have multiple and unpredictable effects" (Gomel 2010, p. 17). Thus, the same event—a fracas at Rudi's restaurant—results in one scenario in which Rudi becomes a smuggler, but another in which he does not. This model of indeterminate and unstable causes and effects at once mirrors and challenges the self-contradictions and alternate histories at the heart of the Brexit campaign. As O'Toole perspicaciously demonstrates, the Second World War constitutes the Jonbar hinge of the Brexit imaginary, constructing a mental space where national and personal identities are configured by binary oppositions: victor and loser, powerful and powerless, and in O'Toole's formulations, "dominant and submissive, colonizer and colonized" (O'Toole 2018, p. 26). As Dave Hutchinson himself put it, "We do a lot of fantastification of our history", cleaving to "this kind of fantasy view of the war" distilled in the iconography of "spitfires over the white cliffs of Dover" (Hutchinson 2021, n.p.). "We don't like to look at history," Hutchinson observes, "we like to colourise it and put cartoon . . . we like to put cat filters on it" (n.p.). "This cartoon version of Britain [ . . . ] does worry me" (n.p.), he admits, drawing a clear line from it to "this cartoon version of Brexit" that touts a return to wartime glory (n.p.), "telling the Germans to go take a running jump" (n.p.).

The *Fractured Europe* quartet accordingly insists that "there are no winners and losers", as a local mafiosi remarks to Rudi (Hutchinson 2016, p. 290), "only survivors" (p. 290). Throughout the books, Rudi gradually learns to accept the unbearable chaos of being (to paraphrase Milan Kundera): "life was like that. It never tied things up neatly" (p. 294). His decisions throughout the first three installments of the quartet are informed by his belief that the smuggling organization of which he forms a part, *Les Coureurs de Bois*, operates as a mysterious illuminati-like cabal that controls his life and gives structure and meaning to his past. However, he eventually discovers, to his disappointment, that *Les Coureurs* is merely an aggregate of freelance operations ensconced in self-generated mystique. This realization prompts an identity crisis that is only partly resolved at the end of the series as he comes to view history with the hard-won understanding that "the world and everything in it had been stupid since the dawn of time" (Hutchinson 2018, p. 357).

On the extra-diegetic level, the *Fractured Europe* quartet enacts the insight articulated by one of its characters, the enigmatic Tomás who specializes in the assessment of holy relics: "History is a strange thing; it keeps being written and rewritten and overwritten and rewritten again" (Hutchinson 2018, p. 175). The *Fractured Europe* books literally rewrite their own history, revisiting the same events and casting new light upon them through fresh points of view. Thus, in *Europe at Midnight*, the storyline of the disillusioned MI5 intelligence officer Jim "Baines" elucidates the events surrounding Rudi's sojourn in London in *Europe in Autumn*. The assassination of a topology expert in *Europe in Winter* is revealed to have been a kidnapping in *Europe at Dawn*, and the decapitated head found in a locker in the first book turns out to have been a bait-and-switch operation in the final book. The intricate temporal relations between these books and within them emerge from Hutchinson's deft interweaving of the epistemological uncertainty of spy fiction and the ontological capaciousness of science fiction. The result is a potent confection that foregrounds the futility of imposing a tidy narrative on the convoluted messiness that is European history. As Rudi concedes with "a great bleakness overtaking his heart" (Hutchinson 2016, p. 267), "History, ... That's a grey area" (p. 267).

Science fiction's capacity to critique the recent rise of right-wing English nationalism extends beyond the subgenre of alternate history. Kevin O'Rourke's *A Short History of Brexit: From Brentry to Backstop* has identified a wide range of factors that contributed to the UK's decision to leave the EU, including the Eurozone crisis, the refugee crisis, the growing socioeconomic inequalities precipitated by globalization, the Austerity measures implemented by the 2010 coalition government and a confluence of chance circumstances and personal decisions (see O'Rourke [2018] 2019, pp. 175–202). Yet, the key factor for the purposes of this article was pinpointed in Maria Sobolewska's and Robert Ford's *Brexitland: Identity, Diversity and the Reshaping of British Politics* (Sobolewska and Ford 2020). Sobolewska and Ford argue that the Brexit vote was strongly driven by "ethnocentrism" (Sobolewska and Ford 2020, p. 22), which they define, following Donald Kinder and Cindy Kam, as "a readiness to reduce society to us *versus* them" (pp. 36–37). In this formulation, Brexit was an act of retrenchment that shored up the bulwark of the "in-group" against perceived threats from the "out-group" (p. 39), the latter framed as "EU immigrants" (p. 226). Thus, the Brexit campaign perpetuated a false dichotomy between Britain, or more precisely England (see Barnett 2017, pp. 100–13), and an imagined entity of "Europeans *en masse*" (Spiering 2014, p. 23). It thereby peddled an image of Europe as "the Other" in Gayatri Chakravorty Spivak's sense of an external, reified object against which the self is defined by way of opposition (see Spivak 1988; Spivak [2003] 2008). Sobolewska and Ford further show that "ethnocentric white school leavers" felt threatened by the ethnic and educational changes that took place in British society throughout the nineties and the first two decades of the twenty-first century (Sobolewska and Ford 2020, p. 36). "From their point of view, it is society that changed and left them behind" (p. 46), as Sobolewska and Ford observe, "Change is perceived as a loss for ethnocentric white voters: a loss of their dominant position, and a loss of the cultural conformity and continuity which they value" (p. 46). In this respect, the Brexit vote was for many voters, if not all, a "momentary stay

against confusion" (Frost [1939] 2007, p. 132), a vote against the frenetic pace of change that has overtaken the twenty-first century.

Yet, no genre has more consistently offered progressive models for grappling with change than science fiction, which Brooks Landon ([1995] 2002) has defined as "the literature of change" (p. xi), elaborating that science fiction "explicitly and self-consciously takes change as its subject and its teleology" (p. xi). As Patrick Parrinder (2000) has argued, science fiction fulfils a social role "by imagining strange worlds" (p. 4), which enable the readers "to see our own conditions of life in a new and potentially revolutionary perspective" (p. 4). Adam Roberts ([2006] 2016) goes further by postulating that "the ur-form of the science fiction text is extraordinary travel, with stories of interplanetary travel the most influential" (p. x). Gary Westfahl (2002) contends rather that "science fiction derives from a more recent awareness that the future might become another distinct world" (p. 2). Whether one privileges interstellar travel or time travel, these studies concur that science fiction is about the encounter with *difference*. In the words of Gregory Benford ([1986] 2018), "There is probably no more fundamental theme in science fiction than the alien" (p. 58). As a genre that perpetually stages the confrontation of the self and the Other, the familiar and the unfamiliar, in an infinite variety of permutations, science fiction is strongly positioned to offer progressive alternatives to the binary logic of Brexit, to challenge and to problematize the Brexit mentality that sets "'us' apart from 'them'" by perpetuating the false dichotomy between England and an imagined "European Other" ((Bauman 2017, p. 50); see (Spiering 2014, pp. 20–29)).

This is not to claim that the entire body of science fiction literature is liberal-progressive. Science fiction novels are often as reactionary, conservative and far-right in their ideological underpinnings as those of any other genre. Tim LaHaye's and Jerry B. Jenkins's *Left Behind* series (LaHaye and Jenkins 1995–2021) is a case in point. Kristian Shaw has identified "a Eurosceptic trend within British fiction" and British science fiction is no exception (Shaw 2018, pp. 18–19). Former conservative MP Edwina Currie's dystopian novel, *The Ambassador* (Currie 1999), feeds directly into English fantasies of the EU as an evil empire intent on world domination. However, a genre as capacious and adaptable as science fiction encompasses Eurosceptic and Europhilic trends alike and a wide spectrum in between. Recent dystopian and borderline-fantastical satirical fiction that grapples with the absurdity of Brexit and the long-term damage it will inflict on both the United Kingdom and Continental Europe includes, inter alia, Sam Byers's *Perfidious Albion* (Byers 2018), Ian McEwan's *The Cockroach* (McEwan 2019), Sibylle Berg's *GRM: Brainfuck* (Berg 2019), and Ali Smith's *Seasonal* quartet (Smith [2016] 2017; Smith 2017, 2019, 2020). Without detracting an iota from the achievements of these novels, the *Fractured Europe* quartet remains the most ambitious attempt to date at science fictional "BrexLit" that tackles the rise of English nationalism and its culmination in the Brexit vote through the topoi, conventions, themes, and narrative structure of the science fiction genre.

## 3. An "Other Place": The Community

The *Fractured Europe* quartet complicates the English Othering of Europe by creating a new "Other place" against which both Europe and England are compelled to define themselves (see Spiering 2014, p. 23). Towards the end of *Europe in Autumn*, the protagonist Rudi discovers the existence of "the Community" (Hutchinson 2014, p. 275), a parallel world "mapped over the top of the whole of Europe and entirely populated by [white-skinned] Englishmen" (p. 275). The Community was created in an alternate nineteenth-century England by a genteel family of cartographers, "the Whitton-Whytes" (p. 265), who discover the secret of creating new worlds by sketching them on survey maps. Thus, the Community is conjured into existence by drawing it over an "Alternative Survey" of the British Isles (p. 269), which subsequent generations of Whitton-Whytes expand to the European Continent and extend "from the Iberian peninsula to a little east of Moscow" (p. 277). This science fictional premise whereby new worlds can literally spring into existence by the act of drawing lines on a map dramatizes and inverts Benedict Anderson's

insight that the nation-state *par excellence* "is an *imagined* political community" (Anderson [1983] 2006, p. 6; emphasis mine). As Henri Lefebvre has shown, the territory of the nation-state is artificially produced by processes of partition, management and control that undergo thorough mystification (see Lefebvre [1975] 2009). In this respect, the act of world building in science fiction is not unlike the constructed reality of nation building. Indeed, one may read the *Fractured Europe* quartet as an exemplar of the ways in which the avowed fictitiousness of the worlds of science fiction calls attention to the obfuscated fictitiousness of nation-states and the borders that define them.

Overtly satirizing postimperial nostalgia, the Community is a "society contentedly stalled in an approximation of the 1950s" (Hutchinson 2016, p. 60), where the only language is English, the only religion is Church of England, and the sartorial fashion is tweed. The result, as attested by one of the protagonists, is appalling:

> Everyone in the Community was English. From one end of the Continent to the other. There were only English things here. There were no other languages, only regional dialects. No other cuisines but English. No other clothing styles but English. No other architectural styles but English. It was awful. (Hutchinson 2015, p. 242)

As this paragraph makes evident, the Community fulfils the isolationist fantasy of right-wing English nationalism and takes it to its logical extreme (see Jeffery et al. 2014). Tom Nairn has traced the regressive impulse evident in present-day conceptualizations of Englishness back to the loss of a distinctive English identity following the dissolution of the British Empire after the Second World War. In the absence of a "coherent, sufficiently democratic myth of Englishness" to replace a sense of imperial belonging (Nairn [1977] 2003, p. 282), xenophobic and racist beliefs became a symptom of identity crisis. These beliefs became forcibly visible in the EU referendum, which was carried by the votes of Englishmen outside of London (see Barnett 2017, p. 101). As outlined in the previous section of this article, many factors contributed to this vote, but scholars widely concur that it was informed by an "English imaginary" in which "England is in and of itself superior to all others" ((Merriman 2018, p. 607); see also (Henderson et al. 2016)).

In a recent interview with the author of this article (February 2021), Dave Hutchinson asserted that "I have always rejected that view of Englishness" that dreams of "building a wall around the coast and saying 'we're gonna be English and Europeans can go away'" (Hutchinson 2021, n.p.). He thus recalled of his developing portrayal of the Community:

> The Community became more Brexit-ey as the books went on. It started out as a lampoon of Little Englishness. That 1950s Ealing comedies, cosy, with working-class people played by upper-class actors trying to put on a cockney accent. . . . it became more Brexit-ey and more ugly. . . . it's a horrible place, which is why in the books a lot of English people want to go there. (n.p.)

Accordingly, far from the epitome of English superiority envisioned by isolationist right-wing nationalists, the Community suffers from technological and cultural stagnation. It has developed nuclear weapons, but not computers, smartphones, or the internet. Its entertainment industry is likewise unexceptional: "In two hundred years, the Community had not provided a single playwright of any great note or a film which would have troubled an Oscar voter for more than a minute" (Hutchinson 2015, p. 242). Nor does it hold the moral high ground. Its ruling body ensures that "dissidence was subtly and totally suppressed and nobody could leave" (Hutchinson 2014, p. 300). Its class stratification is rigid, its labor activists mysteriously disappear from their homes, and a visitor notes that "the sexism here was breathtaking" and racial slurs in common usage (Hutchinson 2016, p. 168). As the books unfold, the protagonists discover that the Community destroyed "the Campus" (Hutchinson 2015, p. 14)—an enormous university campus that occupies the same world as the Community but is cut off from it—and bio-engineered the deadly "Xian Flu" and released it into the population of Europe (Hutchinson 2014, p. 27), resulting in

millions of deaths. By the end of the *Fractured Europe* quartet, the Community has become, in Hutchinson's words, "just like UKIP with nuclear weapons" (Hutchinson 2021, n.p.).

Even as a dystopia, the Community takes on the consistency and hue of porridge, its society as "stodgy and unimaginative and under-spiced" as its culinary range (Hutchinson 2015, p. 242). Its very inception can be read as a process whereby a simulacrum was inscribed upon another simulacrum, the denuded product of a stultified imagination. The Whitton-Whytes model the Community after their "dream of a perfect England" (p. 261), a three-dimensional postcard blown up to enormous proportions, "hundreds of miles, and everywhere was like a garden, laid out with the careful formality of an English landscape gardener" (p. 261). Consequently, while conspiracy theorists cling to the belief that the Community will prove to be "Shangri-La, Utopia, Lyonesse" (Hutchinson 2016, p. 60), they discover to their disappointment that "the Community was *dull*" (p. 60; emphasis in original). Indeed, the very maps on which the Community was created were plagiarized, as the Whitton-Whytes probably stole their measurements from the Ordnance Survey and later copied its sheets (see Hutchinson 2014, pp. 266, 269). Thus, while Oscar Wilde ([1891] 1919) contended that "a map of the world that does not include Utopia is not worth even glancing at" (p. 43), the Whitton-Whytes' map dramatizes the cultural phenomenon that Jameson has diagnosed as "the atrophy of the utopian imagination and of the political vision in our own society" (Jameson [1987] 2005, p. 308). For Jameson, this atrophy is a "pathological symptom of late capitalism" (Jameson [1982] 2005a, p. 345), but a more recent cause may be the corrosion of serious political debate under government administrations whose leaders are better suited to their previous roles as panel-show and game-show hosts. As political discourse denudes into the rhetoric of a tweet or an autocue, it becomes increasingly difficult to summon the energies for rigorous intellectual thought upon which visions of a radically different utopian future are predicated. Far easier to follow the Whitton-Whytes' example, churning out more of the same, endlessly replicating the landscape around their erstwhile estate in Staffordshire.

Notwithstanding that the Community is modelled on a postcard version of England, Rupert, the last survivor of the Campus, is adamant that the Community's population "are *not English*" (Hutchinson 2015, p. 300; emphasis in original). He thus implicitly asserts that England without Europe is no England at all. In Hutchinson's own words, the Community is "England with all the Europe taken out. And it's the Europe which has made us better over the past fifty-odd years" (Hutchinson 2021, n.p.). Thus, by creating the new Other that is the Community, a horrifying ghost of England past, Hutchinson realigns future England with Europe. This realignment is underwritten by Rupert's outsider's perspective as a newly arrived refugee who makes no distinction between England, Britain and Europe in this near-future world. Drifting to the banks of the river Trent and shocked by the deluge of new sights, sounds and smells that assail him in Nottingham, he gives account that "I was overwhelmed by *Europe*" and continues to refer to Nottingham and Europe interchangeably throughout his stay (Hutchinson 2015, p. 117; emphasis in original). Yet, even as he opens a horizon of possibility for England to reimagine its relationship with Europe as one of identification rather than Othering, Hutchinson shows a heightened awareness of England's entrenched Europhobia. As Rupert's handler, the aforementioned MI5 officer Jim "Baines", remarks with bitterness that pays tribute to the influence of John Le Carré: "Imagine a world where there are no *French* people. . . . As far as some of our news organisations are concerned, the Community is the Promised Land" (p. 300).

## 4. Keeping Them at [a] Bay: English Utopian Fiction and Island Mentality

Jim's observation that the Community conforms to the right-wing press's idea of "the Promised Land" lends ironic point to the subtle allusions to utopian literature that pepper the first two books of the *Fractured Europe* quartet. Whilst puzzling over a secret guidebook to the Community, one character speculates that this guidebook must be "a utopian fiction" (Hutchinson 2014, p. 274). This wry suggestion that a world inhabited exclusively by Caucasian Englishmen must be the fictional creation of a utopianist tacitly

alludes to a tradition of English utopian fiction stretching back to Thomas More's ur-text *Utopia* (More [1516] 1923). Lest this allusion go unnoticed, the Campus measures "about two hundred miles across" (Hutchinson 2015, p. 29), a more explicit reference to More's "island of Utopia" that "extends [at its center] for two hundred miles" (More [1516] 1923, p. 41). Hutchinson's evocation of *Utopia* in the context of the Community and the Campus implicitly criticizes the isolationism and xenophobia that underpin many, if not all, invented island utopias in English utopian fiction. As observed by A. L. Morton in *The English Utopia* (Morton 1952), the canonical utopias of English fiction, particularly those set on imaginary islands, often evince "an odious smugness" (p. 9). Gregory Ludlow (1992) has more plainly remarked on the "xenophobic utopian literature of the past" (p. 30). Susan Bruce (2019) has argued of More's "Utopia-the-island" (p. 135): "homogenous in the extreme, insular in every sense of the word . . . monochrome, monoglot, monocultural, monovocal" (p. 135). The people of Francis Bacon's *New Atlantis* (Bacon [1627] 1938) boast in a similar vein of their "rare admission of strangers" (p. 245), and the frame narrator of *The Island of Content* gives account that "our Government will permit us to have no Traffick with any Nation but the *Dutch*" (Ward 1709, p. 29; italics in original), and even then "we keep them at a Bay, . . . and never trust them farther than we are able to manage 'em" (p. 31). Notwithstanding his emigration from the British Isles, Aldous Huxley's *Island* (Huxley 1962) adopted a similar attitude towards its island utopia: "So long as it remains out of touch with the rest of the world, an ideal society can be a viable society" (p. 64). This is not to say that More, Bacon, and Huxley adhered to xenophobic or isolationist worldviews, but rather that an undertow of isolationism and insularity runs beneath their utopian islands regardless of their personal beliefs.

British dystopian fiction expresses a similar concern with the insularity of the British Isles, but rejects its valorization as a precondition for utopia and reconfigures it as a symptom of England's downward spiral under an authoritarian regime. Thus, the dystopias envisioned in George Orwell's *Nineteen Eighty-four* (Orwell [1949] 1998), Alan Moore's and David Lloyd's *V for Vendetta* (Moore and Lloyd 2005), and P. D. James's *The Children of Men* (James 1992) depict an England that is deeply intolerant and xenophobic, scapegoating foreigners and minority groups in order to redirect popular discontent away from its true causes in the policies and actions of the ruling class.

A brief reference to American and Canadian utopian fiction may serve as an instructive counterpoint. William Dean Howells's *The Altrurian Romances* (Howells [1892–1907] 1968), which expressly situate themselves in the tradition of *Utopia* and *New Atlantis* (see Howells [1892–1907] 1968, p. 164), follow their British precursors in asserting that their utopian island of Altruria is "forced to discourage foreign emigration, against their rule of universal hospitality" (p. 273). Austin Tappan Wright's *Islandia* (Wright [1942] 2001), however, is more ambivalent towards the insularity of its titular utopia. Over half of its sprawling narrative revolves around the debate between two political factions over Islandia's future. The progressive party led by Lord Mora seeks to ratify his treaty and open Islandia to international trade and investment, while the conservative party led by Lord Dorn campaigns for the rejection of Mora's treaty and the withdrawal from all but the sparsest foreign relations. Wright gives a fair hearing to both sides, and while the conservatives win the vote, the reader is left with a lingering doubt that "Mora might be right" and prove "the wiser" in the long term (Wright [1942] 2001, p. 530). Matthew Mather's *The Atopia Chronicles* (Mather [2012] 2013) charts the downfall of its titular island of Atopia from a techno-libertarian utopia to a dystopia hijacked by a sadist. Yet, its tragic end notwithstanding, *The Atopia Chronicles* present a serious examination of the utopian ideal of interconnectivity. Atopia embraces existential as well as economic globalization, its cutting-edge technologies allowing the users to transport their consciousnesses from the island to places and times as remote as the twelfth century Mongol invasion, 1980s Los Angeles, the Sahara Desert, and 1920s London. Atopia abolishes physical distance and transcends corporeal limitations, thereby eliminating the preconditions of xenophobia: "nationality was another idea that pssi was about to render irrelevant" (Mather [2012] 2013, p. 369).

The insular mentality expressed in many of the English utopian fictions set on imaginary islands from More's Utopia to Huxley's Island is also apparent in English Euroscepticism (see Spiering 2014, pp. 30–43). The Eurosceptic Norman Tebbit (1990) declared in no uncertain terms: "We must learn how to stay an island. . . . The blessing of insularity has long protected us against rabid dogs and dictators alike" (Tebbit 1990, pp. 76, 78). Tebbit's words and others of similar tenor essentialize the connection between British identity—by which the speaker often means English identity—and an island geography, positing a collective "we" that is defined by being "an island" and an implied European "them" that are fundamentally different because they live on the Continent. Yet, Tebbit's claim that the island geography of the British Isles protects Britons from "dictators", while obviously alluding to the fascist dictators that seized power in Europe in the first half of the twentieth century, glosses over Britain's own imperial history and the inconvenient truth that the British Empire was nothing if not dictatorial, imposing its will across the globe and subjugating millions of people. Political posturing about Brexit discloses a more visible tension between these two mutually exclusive paradigms, insular Britain on the one hand and imperial Britain on the other. Boris Johnson, the current Prime Minister of the UK and one of the leading campaigners for Vote Leave, announced that the UK was "leaving its chrysalis" by leaving the EU (Johnson 2020, n.p.), mixing his metaphors by asserting that Britain was "ready to take off its Clark Kent spectacles and leap into the phone booth and emerge with its cloak flowing" (n.p.). Thus, the UK's post-Brexit future has been envisioned by Brexiteers in the visual iconography of a comic book, whereby butterfly Britain sheds its cover identity and bursts out in full spandex to assume its rightful destiny on the global stage. The puerility of this imagery and its evocation of a superhero whose most iconic superpower is flight unwittingly calls attention to the flightiness of Johnson's vision and its lack of tether to reality. The duality embedded in the figure of Clark Kent-cum-Superman aptly captures the contradictions at the heart of the Brexit vision that superimposes two images of Britain's future as both Clark Kent and Superman at one and the same time. For Johnson and his ilk, Britain is both an insular country that demands from "everybody who comes here and makes their lives here to be and to feel British [ . . . ] And to learn English" (Johnson, qtd. in Barry 2019, n.p.) and the "Singapore of Europe" (Bongiovanni 2018, p. 219), "a global champion" that basks in "the eagerness of our friends around the world to hear once again our independent voice again in free trade negotiations" (Johnson 2020, n.p.). Dave Hutchinson has similarly observed that "what we got with Brexit was a cartoon" that peddles "a vision of England" that allows one to "sit and put your feet up and have a cup of tea" and yet also indulges a fantasy of power and xenophobic violence, "beating the Europeans and thrashing the French" (Hutchinson 2021, n.p.).

The *Fractured Europe* quartet gives the lie to national self-aggrandizement of this stripe. In the twenty-second century envisioned in these books, the Schengen agreement has collapsed, national borders have closed, and European nations have splintered into dozens of ever-smaller polities. As aforementioned, the alternate England of the *Fractured Europe* series is one of the last countries to retain its EU membership, but its fictional trajectory belies the narratives touted by the Leave campaign and the Remain campaign alike. The English are neither colonial subjects of a European superstate nor do they lead the EU into a brighter future. Rather, not unlike many pragmatic Remainers before the EU referendum, they have simply settled into EU membership because "it suits them" (Hutchinson 2016, p. 99). As the EU falls apart in the alternate history of these books, so too does the UK. Far from "beating the Europeans and thrashing the French", the United Kingdom loses its own integrity following the dissolution of the European Union and the balkanization of the Continent, with Scotland and Wales declaring independence and the county of Cornwall in West England attempting to follow suit, sparking "a small civil war" (Hutchinson 2015, p. 122). Hutchinson thereby dramatizes the principle of top-down momentum that governs the logic of communal disintegration once the first step has been taken to assert the primacy of local self-interest over wider collaboration and

compromise. In the alternate Britain of these books, "the truncated United Kingdom" is left "cash-strapped and austere" (p. 72), the Anglo-Scottish border needs defending by garrison patrols, fences, sensors, and drones and London has been wracked by a series of terrorist attacks. Westminster treats its neighbors in the north scarcely better than it treats its neighbors across the channel, refusing "to provide any help" to the struggling Scotts (Hutchinson 2014, p. 231). Newly arrived from the Campus, Rupert's overall impression, once again eliding England, Britain and Europe, is that "England—Britain, Europe, the whole world—had gone through a terrible time" (Hutchinson 2015, p. 123).

## 5. The Science Fictionalization of Europe

In the course of his travels, Rupert contrasts the Community with Europe in highly suggestive terms: "Europe looked like a place from a science fiction novel in comparison" (Hutchinson 2015, p. 240). Hutchinson has commented on this description: "I was trying really hard to see Europe through Rupert's eyes. . . . I just wanted this kind of wide-eyed sense of wonder, I suppose, that he had" (Hutchinson 2021, n.p.). In this endeavor to impart a sense of wonder to Europe, Hutchinson joins a long tradition of English authors who reimagined Europe fantastically.

Europe has been recast as a setting and inspiration for science fiction from the early days of the genre, when English science fiction had just begun to develop a distinct set of conventions in the early- to mid-nineteenth century. According to some genre histories, science fiction was born precisely of the encounter between the English sensibility and the European landscape. Mary Shelley's *Frankenstein* (Shelley [1818] 2020), hailed by Brian Aldiss as the ur-text of Western science fiction (see Aldiss and Wingrove [1973] 1986, pp. 25–52), unfolds on the backdrop of sublime European landscapes, its themes of Promethean ambition and echoing loneliness finding visual correlatives in the snowy mountain peaks, icy glaciers, ruined castles, dark rivers and thick foliage of Europe. The book was famously conceived during Shelley's sojourn on the shores of Lake Geneva while she was reading "volumes of ghost stories, translated from the German into French" (Shelley 1831, p. vii). More broadly, Brian Stableford argues that "the origins of generic speculative fiction are closely connected with the actual development of the steam locomotive" (Stableford 2006, pp. 535–36; see also Adami 2018, pp. 57–64), particularly "the opening of the [English] Stockton-to-Darlington railway in 1825 [that] began a railway revolution that extended throughout Europe" (Stableford 2006, p. 536). Thus, the nineteenth-century railway transport networks that bridged distances between nations on mainland Europe and across the British Isles—and by extension, between the two—inspired and informed the development of modern science fiction. No less a luminary than Charles Dickens (1856) described his railway journey from Paris to London with a markedly science-fictional conceit: "for anything I know, I may be coming from the Moon" (p. 385). For Dickens, the encounter with Europe and especially France was a science-fictional experience that anticipated H. G. Wells's *First Men in the Moon* (pub. 1900–1901). In Dickens's imaginative account, Paris became "the Mooninian capital" (Dickens 1856, p. 386), the Parisians "the Mooninians" who pleasantly frisked about "the Café de la Lune" (p. 385). Nearly a century and a half later, Paul J. McAuley's *Fairyland* (McAuley 1995), a book that Dave Hutchinson has hailed as one of the few British science fiction novels to be "fully engaged with Europe" (Hutchinson 2021, n.p.), made a future permutation of Paris's "old RER line" one of the entrances into the ironically named "Magic Kingdom" (McAuley 1995, p. 153), a derelict theme park on the outskirts of Paris claimed by uplifted genetically engineered humanoids and overflowing with glamour technology.

The *Fractured Europe* books consciously tap into this genre history by investing their utopian energies in an imaginary "TransEurope Rail Route" (Hutchinson 2014, p. 50), nicknamed "the Line" (p. 51), that runs from Portugal to the Chukotka Peninsula (p. 51). Hutchinson explicitly positions the Line as an heir to the Trans-Europe Express, a real railway network that connected Western and Central Europe in the second half of the twentieth century. The first chapter of *Europe in Winter* is titled "Trans-Europe Express"

and depicts a terrorist attack on the Line as shown through the eyes of the perpetrators (Hutchinson 2016, p. 9). By thus foregrounding the Line's debt to the Trans-Europe Express, Hutchinson also alludes to the famous Kraftwerk album and single of the same name (see also Hubble 2017), as well as to the opening sequence of *Fairyland* in which passengers await the departure of a future permutation of the Trans-Europe Express from St Pancras Station (see McAuley 1995, pp. 9–12). Thus, the Line metaphorically draws a line backward from the *Fractured Europe* books to their antecedents, gesturing towards an artistic tradition across media that celebrates the imaginative potentialities of Europe and its transport networks and revels in the creative frisson of English-European encounters.

In its details and idiosyncrasies, the Line recalls Gilles Deleuze's and Félix Guattari's concept of the "rhizome" (see Deleuze and Guattari [1980] 2004, pp. 3–28), a system of connections that resists hierarchical thinking by spreading out from no clearly defined center. The Line's route prioritizes geographic and political exigencies over deference to the established power dynamics within nation-states and between them. Thus, "some polities had two" train stations on the Line (Hutchinson 2014, p. 52), whereas Poland and France each have one, and the Polish station runs past the western border city of Poznań rather than the capital city of Warsaw, to the delight of the Poznanians and the ire of the Polish government. The Line has no central train station, no capital city as it were. Multiple branches seemingly "curl off into nowhere" (p. 276), rendering it a ready-to-hand metaphor for Rudi's reflections about life: "the stories never ended, just branched off into infinity" (Hutchinson 2016, p. 294).

None of this is to say that the *Fractured Europe* books subscribe to the naive ethos of progress epitomized by the railway line in the nineteenth century. Pre-empting such interpretations, the Independent Trans-European Republic that governs the Line soon reveals itself to be as ruthless as one would expect of a secretive government free of democratic accountability. Yet, the Line itself takes on symbolic meaning that transcends its governing body. The material presence of this exquisitely crafted railway that stitches together the fractious nations of Europe provokes inchoate stirrings of hope in Ernst Bloch's sense of the term (see Bloch [1959] 1996, pp. 3–18), particularly among the working-class migrants of this near-future world. A Greek security guard who keeps watch over a stretch of railway in the forests of Sibir perceives it as "a line of light that crossed the whole of Europe" (Hutchinson 2018, p. 285). A Scottish Union Rep of unemployed Polish Metro workers harbors "a sneaking admiration for the Line" (Hutchinson 2016, p. 195):

> He thought that one day the Line might be the only thing holding all those little states together, like the cord running through a string of pearls. (p. 195)

Within the diegesis of this future world, the Line is thereby displaced from the realm of what Lefebvre terms "representations of space" controlled by institutions and their agents to the realm of "representational spaces" that "the imagination seeks to change and appropriate" (Lefebvre [1974] 1991, pp. 38–39). Thus, while the Community is a semi-dystopian simulacrum of England created by drawing lines on a map, the Line is etched upon the physical landscape of Europe and imaginatively reclaimed as a utopian symbol by Europe's migrant workers.

## 6. Encountering the European Self and the European Other

The German philosopher Hans-Georg Gadamer argued that Europe's continued relevance on the global stage derived from "the coexistence of different cultures and languages, religions, and confessions" in Europe (Gadamer [1985] 1992, p. 234), which rendered it "a true training ground" in the hard-won lessons of tolerance and acceptance (p. 234). According to Gadamer, the European condition of longstanding "neighborhood of the other in a narrow space" instilled the understanding that "we are all others and we are all our selves" (p. 234). These insights chime audibly with the growth of Rupert's character, as the only one of the protagonists to have lived in the Campus, the Community and Europe at different stages of his life. Rupert comes to embody an openness to difference, a desire to explore the unknown, and a fluidity of identity that is figured as particularly European (see

also [Hubble 2017](#)). He is drawn to the strangeness, the marvels and the dangers of Europe: "It terrified me and fascinated me, and I wanted to see more" ([Hutchinson 2015](#), p. 117).

Rupert's choice of *nom de guerre*, "Rupert of Hentzau" (p. 13), after the dashing villain of Anthony Hope's Ruritania books, is therefore a highly apposite allusion to the progenitor of a genre of English adventure tales set in an invented European country (see [Daly 2020](#)). The Ruritanian romances foster a fascination with European culture and high society that borders upon the fantastical and dovetails neatly with Rupert's aforementioned impression of Europe as a science fiction landscape. In *Europe at Midnight*, Jim suggests that the name of Hope's English protagonist "would have been [the] more accurate" choice for Rupert's alias ([Hutchinson 2015](#), p. 130). However, Rupert is not merely assuming a false name—he is actively refashioning himself in Europe as a European. The Ruritanian Rupert of Hentzau, who "wandered to and fro over Europe, making a living by his wits" ([Hope 1898](#), p. 2), is therefore the apt choice. True to his namesake, Hutchinson's Rupert embarks on "long hungry tours of the Continent" ([Hutchinson 2016](#), p. 69), exhibiting a similarly Protean selfhood that fashions itself through the landscapes he traverses.

Yet, unlike his namesake, Hutchinson's Rupert develops a compassion for his fellow human beings that transcends the boundaries of nationality and topology. Rupert comes to realize that "while Europe was huge and busy and almost too cosmopolitan to comprehend and full of *potential*" ([Hutchinson 2015](#), pp. 189–90), there was a unifying humanity "under it all" (p. 190): "people were still just people. Angry, decent, captious, stupid, sometimes magnificent" (p. 190). His recognition of a shared European humanity, beyond differences of nation, culture and language, marks him as European in the Gadamerian idealized sense of the term, whereby to be European is to lead "the task of learning to recognize the common in another and in something different" ([Gadamer 1992a](#), p. 219). Indeed, Rupert's willingness to put his life at risk for others coheres with Gadamer's assertion that the future of Europe and the fulfilment of its promise depend on "the risking of one's own [self] for the understanding and recognition of the other" ([Gadamer [1983] 1992](#), p. 207).

Gadamer sounded a cautionary note regarding twentieth-century technologies, which he believed posed a risk to the cultural diversity of Europe by facilitating homogenization and hamstringing identity formation and intellectual and creative maturation (see [Gadamer [1983] 1992](#), p. 202). *The Fractured Europe* books promulgate the opposite view, illustrating Roger Luckhurst's contention that science fiction is intensely "focused on the fantastic enablements and disablements of the subject in technological modernity" ([Luckhurst 2010](#), p. 11). Far from alienating the subject from itself and diluting cultural specificities, in *Europe at Dawn* technology takes on the symbolic meaning of "technologies of the self" ([Foucault 1988](#), p. 17), a term used by Michel Foucault to denote the techniques by which individuals shape "their own bodies and souls, thoughts, conduct, and way of being" (p. 18). The digital camera becomes a catalyst for creativity in the hands of Alice, a Scottish diplomat-turned-painter first introduced in *Europe at Dawn*. Alice is forced into exile in a remote Estonian village, where she lives under a false name and is cut off from all meaningful human connections. She thus initially feels alienated from her environment and creatively sterile: "She'd soldiered on, standing at her easel in a field somewhere or on a hill or in a wood, and everything she painted was terrible" ([Hutchinson 2018](#), p. 319). Yet, her creative energies are restored through the mediation of photography: "It wasn't until she tried working from a photograph that she produced anything remotely worthwhile" (p. 319).

The special role that Hutchinson accords to photography in this scene recalls Walter Benjamin's claim that the photograph displaces the object of representation—"The cathedral leaves its locale to be received in the studio of a lover of art" ([Benjamin [1935] 1968](#), p. 221)—but Hutchinson takes this idea further than Benjamin intended. Not only does photography bring the European landscape into Alice's studio, but it enframes that landscape and makes it legible. Her European environment can only become creatively meaningful to her when captured through the lens of a camera, blown up and printed, enabling her to paint so long as she positions herself at a double remove from the object of representation.

This process is facilitated through Alice's tactile connection with old-fashioned technology, as she prefers "an antique digital camera she'd found in a junk shop" to her iPad and she considers "getting hold of an old analogue camera and trying that" (Hutchinson 2018, p. 319). Given that Alice's digital camera is considered an antique in this future world, her fondness for it resonates with a wider penchant for the obsolete curiosity in British fantastical fiction. Hutchinson thereby expresses an ambivalence towards what Gadamer regarded as "the tempo of the technological age" (Gadamer 1992b, p. 137), an ambivalence shared by many authors of British and European speculative fiction that often contrasts with the more technophilic tenor of their American counterparts (see Elber-Aviram 2021, p. 108).

Painting the European landscape from photographs enables the Scottish Alice to glean insight into the European other, namely the Estonian Rudi, and into his relationship to this environment. Rudi is a stranger to Alice at this stage of the story, but by painting from "dozens of photos" (Hutchinson 2018, p. 319), two of which capture him "walking across the middle distance" (p. 320), she penetrates the picturesque surface to capture a deeper truth about his loneliness and inner strength:

> A couple of brush strokes, no more, and there he was, a tiny figure lost in a huge landscape with the whole world behind him. She thought she detected a certain determination in the way she'd painted him. (p. 320)

The referent described by the phrase "a certain determination" is tellingly ambiguous—it could refer equally to Rudi's portrayal and to Alice's artistic style. Thus, the boundary between the Scottish painter and the painted Estonian becomes blurred, self and other ambiguously confronted in an extended tableau that forms an ekphrastic interruption within the text. Small wonder that by the end of the process Alice's thoughts about the painting become increasingly muddled: "This one wasn't bad. Certainly not bad enough not to sell. She'd done much worse" (p. 321).

## 7. Beyond Utopia and Dystopia: Cooking in a Kraków Kitchen

Together with photography, the *Fractured Europe* quartet valorizes cooking as a privileged "technology of the self" that opens a space for identity formation within the infinite complexity of modern Europe (Foucault 1988, p. 45). Hutchinson has observed that "people don't eat enough in science fiction" (Hutchinson 2021, n.p.), an omission that he has strived to redress in the *Fractured Europe* quartet: "I thought, well, if nobody else does it, at least I'll have people eating in my books" (n.p.), so that "at least . . . in one science fiction novel people will have a decent meal" (n.p.). Hutchinson's future Europe thus offers a wide range of cuisines, showcasing the creativity and diversity of the European palate, from "trays of anchovies and chopped onions" (Hutchinson 2014, p. 14), to steak tartars made of "minced beef, each with an egg yolk nestling in a hollow on top" (p. 14), to "pork cutlet and a potato pancake" (p. 50), chips served in "small cowboy hats" (Hutchinson 2016, p. 36), "poached pears for dessert" (Hutchinson 2018, p. 246), and "a small plate of amaretti" (p. 247). This culinary scene assumes a transnational character, with Estonian chefs cooking in Poland, Norwegian chefs cooking in Estonia, Turkish chefs cooking in Latvia, MI5 officers eating out at a Polish restaurant in London, and Rudi dining with Rupert at a Brazilian restaurant in Prague. The Community's culinary offerings literally pale in comparison, so much so that Rupert "would gladly have lynched someone for a kebab" after living there for a year (Hutchinson 2015, p. 242). The head of the Community's intelligence services evidently agrees—he sends Rupert on a covert mission to London to smuggle out a large Fortnum & Mason picnic hamper. He is later seen savoring *foie gras* and caviar on biscuits. "We're English. Beer drinkers to a man, none of that foreign muck" (Hutchinson 2016, p. 164), he remarks as he pours himself a glass of Fortnum & Mason red wine.

Michel de Certeau ([1980] 1984) has recognized "the art of cooking" as a major tactic for "establishing a kind of reliability within the situations imposed on an individual" in an expansive technocratic society (p. xxii). Certeau's collaborator Luce Giard ([1980] 1998) further argues that cooking domesticates space—it is "a way of being-in-the-world and

making it one's home" (p. 154). This paradigm coheres with Rudi's work as a chef in a restaurant in Kraków that he comes to regard as his home, "the safe space" to which he returns when he is "feeling weirdly adrift all of a sudden" (Hutchinson 2016, pp. 69, 68). Notwithstanding that the books follow his adventures as a smuggler working for the mysterious organization *Les Coureurs de Bois*, Rudi self-identifies as "a chef" rather than a Coureur ((Hutchinson 2016, p. 71); see also (Hutchinson 2014, p. 45)). He is first introduced "standing in the kitchen doorway" of the restaurant and three of the four books end with Rudi offering to cook a meal, about to prepare one or contemplating doing so (Hutchinson 2014, p. 10; see Hutchinson 2015, p. 303; Hutchinson 2016, p. 295; Hutchinson 2018, p. 362).

This figuration of the restaurant as Rudi's home is crucially welded to a cosmopolitan vision. Hutchinson recalled that the restaurant was envisaged as "Europe in microcosm" (Hutchinson 2021, n.p.): "a bunch of Hungarians come into a restaurant in Poland, owned by an ethnic Silesian, with an Estonian chef. I wanted that sense of a cosmopolitan Continent right from the start" (Hutchinson and Pintér 2017, n.p.). Its cosmopolitan character emerges in full at the end of *Europe in Winter*, as Rudi looks around the restaurant:

> It looked like the setting for a particularly complicated joke. *People of many nationalities walk into a bar . . .* There were Poles here, Silesians, Kosovars, Italian and French chefs, a Spanish restaurateur, the senior staff from Max's Berlin restaurant, two English people, an Estonian, and the last citizen of the Campus. (Hutchinson 2016, p. 293)

This depiction of the restaurant as cosmopolitan and domestic, homely and pan-European, renders it a locus of Gadamer's vision of European diversity that teaches the world "to learn to live with others" (Gadamer [1985] 1992, p. 234). The restaurant, and by extension other places in which Rudi cooks, becomes an enclave of tolerance and restraint. In the opening sequence of *Europe in Autumn*, a group of rowdy Hungarians create a ruckus at the restaurant, but the owner serves their courses with unshakeable grace, reasoning that were he to call on the services of the local protection racket, "half of us would have wound up in the mortuary" (Hutchinson 2014, p. 12). Rudi follows suit after he inherits the restaurant, as the Hungarians, who have since become his allies, offer to kneecap a man who has been threatening his life: "For a moment, the thought did appeal, but Rudi shook his head" (Hutchinson 2016, p. 295). In similar vein, Rudi half-heartedly suggests killing a woman who has threatened to drop dirty bombs on Europe and possesses the means of doing so. However, he quickly dismisses this notion from his mind and elects rather to take charge of the outdoors grill:

> He found himself gravitating, by default, towards the barbecues, and shortly after that he was cooking burgers and sausages and chicken drumsticks and handing them out on paper plates to whoever went past. Which was obviously what one did when one had tracked a supervillain down to their secret headquarters and exposed their evil plan for world domination. (Hutchinson 2018, p. 358)

For all its self-deprecation, Rudi's decision to cook rather than kill carries a serious ethical point. It incubates what Caroline Edwards (2019), following Ernst Bloch, has described as "utopian anticipatory consciousness, uncovered through the mundane facets of daily life and everyday behaviours" (p. 26). The *Fractured Europe* books suggest that to take responsibility for the other's culinary needs is the first step towards a utopian horizon where the self has learned how "to live with an other, to live as the other of the other" (Gadamer [1985] 1992, p. 234).

At the end of the *Fractured Europe* quartet, Rudi finds the sought-after mathematical equations that enable the manipulation of existing worlds and the creation of new ones. He thus gains the power of a god, or indeed of a science fiction writer, to conjure up an entirely new world or remake an existing one in his own image. Yet, Hutchinson takes pains to render this key moment anticlimactic. The mathematical equations are written on a sheaf of papers that constitute one copy among many, and they have been overlaid by the crayon

scribblings of a small child. These scribbles create a palimpsest that recalls the cartographic experiments of the Whitton-Whytes, but where the latter drew lines on a plagiarized map of England to create a new world, the child scrawls "animals and houses and people" over "the arcane mathematics" (Hutchinson 2018, p. 361), "diligently attempting to conceal" them (p. 361). She thus obscures the secret of creating new worlds with imaginative drawings of her own environment, sketches that are significantly uninflected by nation, race, class, culture, gender, or even animal species: "Her horses were pretty good. Or maybe they were dogs" (p. 361).

Upon discovering this sheaf, Rudi is briefly tempted "to start writing and rewriting universes" (p. 361). However, on further consideration, he sides with the scribbling child over the cartographers with utopian aspirations. He literally casts the sheaf away, heeding Rupert's advice: "Go home, cook food" (p. 358). Rudi thereby elects to "stay with the trouble" in Donna Haraway's phrase (Haraway 2016, p. 2), to live and cook in an imperfect world rather than escape into a perfect one of his own devising. In so doing, he shoulders his ethical responsibility, taking "up a pair of tongs" and flipping burgers rather than dispensing with the labor by creating "a universe without" them (Hutchinson 2018, pp. 359, 361). "And yes, I'm aware it's not enough" (p. 359), he replies to Rupert's skepticism, "but I don't have anything else" (p. 359). Beyond utopia and dystopia, Rudi and the reader are left with a splintered Europe on the cusp of change, a painting made from photographs, the inchoate squiggles of a young girl, and the enticing smell of food wafting in from the kitchen.

**Funding:** This research received no external funding.

**Acknowledgments:** My heartfelt gratitude to Dave Hutchinson, who gave his time to a three-hour interview with extraordinary openness, honesty, depth of insight and generosity of spirit. I am deeply grateful to Elana Gomel, who invited this contribution to her Special Issue, provided perspicacious feedback and encouraged and supported it with infinite patience. I owe a great debt to Simon Bacon, who read an early draft of this article and provided invaluable feedback on it. My thanks to the Humanities Editorial Office, and especially Kai iizuka, who showed great patience, empathy and understanding as revisions to this article were delayed due to the pandemic. I also profited greatly from the input and suggestions of the anonymous peer-reviewers, for which they have my thanks. Additional thanks are due to Nick Hubble, who first introduced me to Hutchinson's Fractured Europe quartet, and to Jim Clarke, Andrew M. Butler, Adela Terrell, Edward James, Paul March-Russell and Jo Lindsay Walton of the London Science Fiction Research Community for sharing their expertise, connections, kindness and enthusiasm in support of this article.

**Conflicts of Interest:** The author declares no conflict of interest.

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
