# Peer review of "Rewriting Universes: Post-Brexit Futures in Dave Hutchinson’s Fractured Europe Quartet"

_humanities, doi:10.3390/h10030100_

Round 1
Reviewer 1 Report
Fascinating and well-informed article: original, rigorous and quite significant. Recommend publication.
Author Response
Thank you so much to this reviewer for their extremely kind and generous words. Kindness and generosity are rare qualities in peer-reviewers, and I would like to thank them sincerely and deeply. Please convey my wholehearted gratitude, and please tell them that I am truly touched.
Reviewer 2 Report
The manuscript examines the literary response to Brexit focusing on Dave Hutchinson’s science fiction Fractured Europe quartet of novels about the disintegration of a Europe located in the near future. The essay ‘grapples with the causes and consequences of the United Kingdom’s vote to leave the European Union’, but from a rather one sided perspective. Building on work, most notably the Robert Eaglestone 2018 edited volume Brexit and Literature ,on this new genre, this article shifts the focus from literary fiction to science fiction.
The manuscript uncritically adopts and reiterates the critical literary position outlined in the Eaglestone volume that the vote to leave the European Union in 2016 was an unprecedented and unthinking act of self -harm. It adumbrates the Eaglestone volume with an additional layer of postmodernist critical thinking to suggest that proponents of Brexit cynically set out a ‘jingoist’ agenda to ‘other’ Europe and cynically engineer a return to a narrow tribal and nationalist understanding that contrasts diametrically with the progressive values associated with the European project.
Hutchinson’s quartet, like most Brexlit (to use Christian Shaw’s felicitous term), makes the assumption, as does a wider English literature academic orthodoxy, that anyone who voted for Brexit is a white racist, lost in an atavistic nostalgia for an imperial past. At the same time, all those who voted Remain, that includes inter alia the humanities departments of most universities, the mainstream media and most publishing houses are both Europhile, enlightened and morally superior cosmopolitan idealists. This simplistic Manichean dualism pervades what the author considers Eaglestone and his contributors seminal volume on Brexit and Literature, as well as the novels written in this genre ranging from inter alia Ali Smith’s Autumn Rachel Cusk’s Kudos and Jonathan Coe’s Middle England to Dave Hutchinson's overwrought quartet -the subject of this manuscript under review. The manuscript nowhere discusses those novels , apart from Hutchinson’s, that address the Brexit topoi or the predictably Remainer line they advance. Given that Hutchinson’s work uses science fiction and satire to examine Brexit and the English relationship to Europe it is surprising the manuscript fails to refer to other works in this genre like Douglas Board’s Time of Lies, Sam Byers' Perfidious Albion or Ian McEwan’s Cockroach both to indicate similarities in themes pursued and for purposes of comparison and contrast.
Ultimately, I found little in this essay that added to what we know that academe and the contemporary British literary establishment thinks and writes about Brexit. The author’s grasp of English satire in a science fiction or futurist idiom also seems a little shaky. Following A.L. Morton’s (1952) captious and somewhat outdated view of English utopian fantasy as characterised by an ‘odious smugness’ the author anachronistically condemns both Francis Bacon’s The New Atlantis and Thomas More’s Utopia and with a total disregard for context. The ‘monochrome’ More, the author might be surprised to learn, died for the cause of European Christendom against the first attempt at creating a separate English ‘Empire’ (see the preamble to the Act of Succession 1534). In a similar partisan vein the author ignores the great works of English political satire from Swift’s Gulliver’s Travels, to Orwell’s 1984 and Huxley’s Brave New World presumably because they do not fit the author’s assumption of a monochrome and smug xenophobia permeating English satirical literature prior to the emergence of Brexlit.
The strength of the great French and English nineteenth century novelists was their attempt to understand the motivation of those whose actions they may have otherwise deplored. They tried to explore character rather than offer crude stereotypes. Brexlit and its admirers, by contrast have replaced imaginative empathy with hysterical, ideological denunciation and a rather disturbing version of conspiracy theory.
Any thoughtful interrogation of this genre should surely now give some attention to those who might very much like the idea of Europe in the sense that Gadamer or Zizek advocates, but find , as evidenced by the EU's recent vaccine programme and its pursuit of a single currency, a recipe for European dissolution rather than ever closer union. Unfortunately, this manuscript only serves to reinforce a fashionable and predictable Brexlit orthodoxy.
Author Response
I would like to thank the peer-reviewer for their input and suggestions. Following their review, I made the following revisions:
- I added a brief contextualisation of other works of BrexLit, both Eurosceptic and Europhilic. See page five of the revised manuscript, lines 228-241.
- I added qualifications to my discussion of English utopian fiction and I clarified that insularity and xenophobia are apparent in the work of More, Bacon, and Huxley, quite regardless of their personal beliefs and conduct. See page eight of the revised manuscript, lines 368, 370-1, 382-385.
- I added a brief reference to English dystopian fiction that addresses similar themes in a more critical light. See page eight of the revised manuscript, lines 386-393.
Reviewer 3 Report
I'd recommend a bit of revision to the introductory section on Brexit.
This is a strong essay on a quartet of novels that has not been previously worked on by science fiction critics. It deploys a useful interview with the novelist, and it picks out some crucial components of how a science-fictional approach to Englishness and Brexit can begin to offer a critique of the right-wing nationalist reaction to the EU after 2016. This material is strong, and seems to me in a good state as it stands. It's quite narrowly focused (there are no other major SF works of the same period alongside it, really -- if some insightful background from the 1990s, via McAuley). The four books rest in isolation -- even in Hutchinson's larger body of work (he has subsequently contributed to the dystopian genre with *Shelter*). But this narrow focus is okay -- there can be a benefit in keeping this targeted on the quartet.
The main issue that does need some address is the rather underpowered nature of the discussion of Brexit itself. There is no real discussion of the shifting debate between 2016 and eventual exit in 2021, nor any sense of the history of how that decision was arrived at, nor the subtle and often quite divergent positions held on both sides of the debate. The essay opts for a rather rudimentary sense that Brexit came from 'Othering' Europe. This seems to simplify. There were lots of contradictions here, even on the pro-Brexit right (for example, the appeal to closed borders and end of free movement of people, alongside the fervent wish for continued free movement of capital, or the tying together of a 'Little Englander' position unstably with a 'Global Britain' vision of the UK as a new Singapore -- often promoted by the same voices). I think the instability and contradictions of Brexit discourse need more acknowledgement -- not least because I think this fosters Hutchinson's vision also of multiple/alternate worlds/pocket universes of the quartet. The white-only Community is a certain version of UKIP's offer, but there were other, contradictory versions on offer too, which is why it was for a very long time in paralysis or stasis under Theresa May, who could not reconcile these divergent forces. It's that stasis, and also the thin possibility of abandoning the whole project, that kept the UK in thrall for those 5 years. I'd recommend for history, Kevin O'Rourke's A Short History of Brexit, and for the contradictions at the heart of Brexit discourse Fintan O'Toole's Heroic Failure (the best single book on Brexit so far, I think). This would help beef up the introductory section and take it into a more nuanced position, where Self/Other divides can work in multiple ways, not just a negative othering, before heading into the Hutchinson sections which work well at the moment. The recent sociological study, Brexitland, also does something to break open the sense that one form of reactionary nationalism was alone responsible for the vote (Sobolewska and Ford).
Author Response
Thank you so much to this peer-reviewer for their erudite, in-depth, and extremely helpful review. Their suggestions on how to improve and expand the section on Brexit have been invaluable, and my discussion of Brexit has improved significantly as a result. Please convey my heartfelt thanks!
Following their review, I have expanded and deepened the sections on Brexit considerably. I have drawn extensively on Fintan O'Toole's Heroic Failure (I agree with the peer-reviewer, it's a great book) and more briefly on Kevin O'Rourke's A Short History of Brexit and Maria Sobolewska and Robert Ford's Brexitland: Identity, Diversity and the Reshaping of British Politics.
Please see pages 2-5 and 9 in the revised manuscript.